# The Burden of Urinary Tract Infections on Quality of Life and Healthcare in Patients with Interstitial Cystitis

**DOI:** 10.3390/healthcare11202761

**Published:** 2023-10-18

**Authors:** Cléo Baars, Charlotte van Ginkel, John Heesakkers, Mathilde Scholtes, Frank Martens, Dick Janssen

**Affiliations:** 1Department of Urology, Radboudumc, 6525 GA Nijmegen, The Netherlands; cleo.baars@radboudumc.nl (C.B.); frank.martens@radboudumc.nl (F.M.); dick.janssen@radboudumc.nl (D.J.); 2Department of Urology, Maastricht UMC+, 6229 HX Maastricht, The Netherlands; john.heesakkers@mumc.nl; 3Interstitial Cystitis Patient Association ICP, 4000 AB Tiel, The Netherlands; mjmjscholtes@icloud.com

**Keywords:** interstitial cystitis, bladder pain syndrome, urinary tract infections, burden of illness, quality of life, healthcare surveys

## Abstract

Background: Interstitial cystitis/bladder pain syndrome (IC/BPS) patients are more susceptible to urinary tract infections (UTIs), likely worsening pre-existing symptoms. However, this receives limited attention in guidelines. This study aimed to explore the burden of UTIs on IC/BPS patients’ quality of life and their healthcare. Methods: Two quantitative retrospective database studies were conducted in cystoscopically proven IC/BPS patients: one compiled existing patient survey data (n = 217) from July 2021 to examine physical and emotional UTI burden, and the other used a patient file database (n = 100) from January 2020 to May 2022, focusing on UTI prevalence, healthcare use, urine cultures and antibiotic resistance. Results: A delay in diagnosis was seen in 70% of patients, due to doctors confusing IC/BPS symptoms with UTIs. The UTI prevalence was over 50%; these patients also report anxiety for new UTIs (70%) and worsening of IC/BPS symptoms (60%). Additionally, for UTI+ patients, healthcare consumption was significantly increased in both studies. Antibiotic resistance (80% of cultures) and prophylactic antibiotic use were common. Conclusions: These findings highlight the burden of UTIs on quality of life in IC/BPS patients and the healthcare system. These results emphasize the need for improved UTI guidelines concerning diagnosis, management and prevention for IC/BPS patients to improve quality of life and care.

## 1. Introduction

Interstitial cystitis/bladder pain syndrome (IC/BPS) is a chronic inflammatory bladder disease, defined as “chronic pain/discomfort in the pelvic region related to the urinary bladder (>6 months), accompanied by at least one of the following: urgency or frequency” [1]. It is a symptom-based diagnosis, based on the exclusion of other identifiable diseases, e.g., urinary tract infections (UTI) or carcinoma in situ [1]. The International Society for the Study of Bladder Pain Syndrome (ESSIC) defines three subtypes of IC/BPS; types 1, 2 and 3, cystoscopically identified as, respectively, no abnormalities, glomerulations and typical red lesions called Hunner Lesions [1]. The underlying pathophysiology is unknown, but the association is made with a disruption of the urothelial barrier and an increased permeability, which leads to inflammation in the underlaying layers [2].

Patients with IC/BPS often present at the urologist with a medical history of UTI-like symptoms, oftentimes with negative urine cultures; therefore, UTI is considered a confusable disease for IC/BPS. Many patients still receive frequent courses of antibiotics and there is an expected diagnostic delay. However, UTIs might also coincide with IC/BPS. The patients are more susceptible to UTIs; this is believed to be caused by easier bacterial adherence to the bladder wall due to an impaired mucosal barrier [3,4]. IC/BPS and recurrent UTIs can both negatively affect quality of life (QoL) [5,6,7]. Also, both groups are heavy healthcare consumers. Analyses of QoL and healthcare consumption has been performed for IC/BPS and recurrent UTI separately [8,9,10,11]. However, there is a need to zoom in on the specific IC/BPS group with UTIs, because UTIs are likely to negatively impact pre-existing IC/BPS symptoms. Nonetheless, there is little attention in the literature and guidelines given to the presence, burden and management of UTIs in these patients [12,13,14]. The relation between IC/BPS and UTIs, and the associated clinical implications on healthcare, have been poorly explored in research. Currently, it is not well-known how many or which IC/BPS patients are affected by UTIs. More insight into this topic could improve healthcare management in IC/BPS. 

The aim of this study was to clarify the burden of UTIs on IC/BPS patients’ quality of life and the healthcare system by using patient reported data from a Dutch survey combined with real-world clinical data from four high-volume medical centers in the Netherlands.

## 2. Materials and Methods

### 2.1. Methodology and Ethical Statements

The burden of UTIs on IC/BPS patients and their healthcare consumption was investigated by combining the patient (subjective) perspective with the (objective) data from patient files. The patient perspective was assessed through a quantitative database study. The database was made available by the Dutch IC/BPS patient association (ICP) and composed of survey results that were previously collected by the ICP. This database study received approval from our institute under registration number: 115001. No ethical consent from the Ethical Review Board (Centrale Commissie Mensgebonden Onderzoek Oost Nederland) was needed because of the non-invasive and non-burdensome nature of the study. The patient file study consisted of retrospective data collection and was conducted in four healthcare centers: Radboud university medical center, Andros clinics Baarn, Andros clinics Arnhem and Andros clinics Den Haag. The patient file study received approval from our institute under registration number 114972. No ethical consent from the Ethical Review Board (Centrale Commissie Mensgebonden Onderzoek Oost Nederland) was needed because of the non-invasive and non-burdensome nature of the study.

### 2.2. Study Design

Firstly, the database included answers to 147 questions on symptoms and QoL filled out voluntarily by IC/BPS patients who were associated with the ICP. The ICP collected these data anonymously in July 2021; 217 participants responded, with a reported response rate of 48%. The database included individuals diagnosed with IC/BPS, former IC/BPS patients and partners. Adult patients who had an IC/BPS diagnosis at the time of the questionnaire were included for analysis. Excluding former patients and partners, additionally, IC/BPS patients who had undergone cystectomy were excluded from analysis. The main study parameters in this study were the presence of UTIs and the influence of UTIs on the physical and emotional wellbeing of IC/BPS patients, with the latter including diagnosis delay, complaint rates after UTI and fear of developing UTIs. The following secondary parameters were collected: use of prophylactic and therapeutic antimicrobial treatment, urinary cultures (including antibiotic resistance) and reported healthcare consumption for IC/BPS due to UTIs. Patients’ characteristics, such as age, sex, subtype of IC/BPS and presence of Hunner lesions, were analyzed as well. A section of 18 questions were used for this study to measure the abovementioned study parameters, starting with multiple-choice questions concerning the diagnosis, duration of complaints, delay in diagnosis and performed cystoscopy. This was followed by several multiple-choice questions concerning the presence of UTI, diagnosis and antibiotic use. A visual analogue scale was used to ask about the negative influence of UTIs on the symptoms of IC/BPS, from 0 (no influence) to 10 (very strong influence). This section ended with three questions that asked the participants to rate how strongly they agreed with statements relating to the impact of UTIs on the symptoms and course of IC/BPS. Answers were on a five-point Likert scale. In analysis, the answers ‘agree’ and ‘moderately agree’ were seen as agreeing with the statement; the answers ‘unclear’, ‘moderately disagree’ and ‘disagree’ were seen as not agreeing with the statement. The final two questions were multiple choice regarding sex and age. The questions are attached in Appendix A. 

Secondly, the patient file study was conducted in 100 randomly selected adult patients who were registered under diagnosis code ‘Interstitial cystitis’ and who were followed-up between January 2020 and May 2022. The data were processed anonymously from electronic patient files from four healthcare centers in the Netherlands specializing in IC/BPS healthcare. For inclusion, diagnosis had to have been made with cystoscopy by a urologist. Urine sediment or dip slide check (to rule out a UTI by absence of nitrite) prior to cystoscopy was standard protocol in all study sites. Patients were labeled as either Hunner lesion positive (HL+) or Hunner lesion negative (HL−) based on cystoscopy. Criteria for excluding subjects were as follows: no cystoscopy performed and participation in an intervention trial for treatment of IC/BPS. The main study parameter was the number of UTIs per year per patient (anamnestic in patient interview, positive culture or nitrate-positive urine stick). This was followed by healthcare consumption, which was measured through number of contacts, visits and hospital admissions to the department of urology. Demographic variables such as sex and age were gathered. Moreover, secondary parameters included antibiotic use, prophylactic-treatment use (glycosaminoglycan (GAG) therapy, antibiotics, cranberry products) and urinary cultures, including their antibiogram.

### 2.3. Statistical Analysis

The data were analyzed using descriptive statistics; valid percentages were used. For further analysis, patients were divided into different groups. For the survey database, the patients were divided into history with UTIs and without UTIs. For the patient file study, firstly, patients were divided into patients with and without UTIs. Moreover, the subgroups HL+ and HL− were used to evaluate the differences between these subtypes. Data were analyzed using the Man-Whitney U test and the Chi square test according to the type of variable. Statistical significance was set at 5%.

## 3. Results

### 3.1. Demographics

The age and sex distribution in both patient cohorts was quite similar, aligning with the well-established higher prevalence of IC/BPS in females. In the patient file study, there was a noticeably higher prevalence of HL (96%), whereas in the survey database this was evidently lower (46%). Most patients had symptoms long before diagnosis. Just over half of the surveyed participants reported symptoms in the 2 years prior to diagnosis, and almost a quarter reported having symptoms for over 10 years. In both studies, we found that men had significantly fewer urinary tract infections. The demographic data are shown in Table 1.

### 3.2. Patient Survey Database

Data were collected from 217 individuals, of whom 198 had symptoms of IC/BPS at the time of the questionnaire and were included for analysis (Table 2). Seventy percent of the patients reported that the general practitioner had confused their symptoms with UTIs, leading to delayed diagnoses of IC/BPS for most patients. This misidentification was significantly more prevalent among patients experiencing concurrent UTIs, with rates of 84% compared with 54% in those without concurrent UTIs (*p* ≤ 0.001). However, it did not result in a longer delay in diagnosis. 

Just over half (55%) of the participants reported having current or prior UTI problems. Of these patients, the majority needed one or more therapeutic antibiotic treatments in the past two years, with some individuals (21%) receiving as many as six or more courses of antibiotics. Concerning prophylactic use of antibiotics, half of the participants received this type of treatment. Moreover, 32% indicated resistance to antibiotics in their urine cultures. 

Sixty percent of the participants with UTIs reported that their IC/BPS symptoms had worsened severely after a UTI. When the surveyed were asked how much UTIs negatively affect their IC/BPS symptoms on a VAS scale from 0 to 10, the median rating was 7.0 (IQR 6.0). Subsequently, the majority of the patients experienced anxiety about developing a new UTI. Not surprisingly, looking in detail at patients who experienced concurrent UTIs, these numbers were significantly higher for these patients in comparison with IC/BPS patients not reporting issues with UTIs. Furthermore, an increase in healthcare utilization for their IC/BPS during or after a UTI was more commonly reported in cases of (recurrent) UTIs compared with isolated IC/BPS.

### 3.3. Patient Files Database

In total, 163 electronic patient files of IC/BPS patients were randomly screened until 100 patient files that met the inclusion criteria were included and further analyzed for the period of January 2020–May 2022. The summarized data are shown in Table 3.

Sixty-three percent of the patients had a history of UTIs, established during patient interviews, and 53% of the patients had a UTI between 2020 and 2022 (UTI+), with a median number of almost one UTI per year. UTI+ patients utilized significantly more urologic care in comparison with UTI− patients, with respective medians for hospital contacts of 7.2 (IQR 9.6) and 4.4 (IQR 6.0) per year (*p* = 0.007). When analyzed in more detail concerning different types of hospital contacts (digital, telephone and physical), all differed significantly between these groups, with the most substantial disparity observed in the number of physical consultations: 15 (IQR 17) for UTI+ and 9 (IQR 12) for UTI−. There was no significant difference in the number of hospital contacts (digital, telephone or physical) between HL+ and HL− patients. 

Not surprisingly, patients experiencing UTIs received significantly more therapeutic and prophylactic antibiotic courses. For prophylactic antibiotic use, 21% of patients who did not experience a UTI throughout the study period did continue to use these antibiotics consistently throughout that same time frame. GAG therapy was also used commonly, with only a slight increase in use in the UTI+ group (85% vs. 72%). 

Remarkably, only 28% of patients had a urine culture taken in the study period, with a median number of 2 (range 1–18). Of these patients 75% had at least one positive culture. Commonly cultured pathogens were: Escheria Coli (32%), the Klebsiella family (25%), Proteus mirabillis (9%), yeasts (17%) and others (17%). In most cases (86%), monopathogens were cultured. 

Eighty percent of patients with positive urine cultures showed antibiotic resistance in these cultures. Resistance to amoxicillin (clavulanic acid) was most common (11 patients), followed by trimethoprim/co-trimoxazole (7 patients) and fosfomycin (7 patients). 

## 4. Discussion

This analysis shows that UTIs are common in this IC/BPS patient cohort, with a prevalence of over 50% (median of almost one UTI per year). UTIs place a burden on QoL, which consists of the increase in IC/BPS complaints after a UTI, the fear of developing new UTIs and the delay in diagnoses caused by doctors mistaking IC/BPS symptoms for symptoms of a UTI. The second major finding is the burden on the healthcare system. Firstly, this was because patients indicated an increased healthcare consumption after a UTI. Secondly, UTI+ patients had significantly more hospital visits for their IC/BPS symptoms (not specifically for UTI flares) compared with patients without UTIs.

The aim of the study was to assess the burden of UTIs on IC/BPS patients. UTIs are seen as a confusable disease for IC/BPS, but our study shows they also frequently occur concordantly. The hypothesis of the altered urothelial barrier and dysfunctional voiding patterns could be an explanation for the high occurrence of UTIs in IC/BPS patients. Expert opinions often state the high prevalence of UTIs; however, evidence on this is limited. Previous studies found varying results and lower percentages than in our patient cohort. Nickel et al. found a prevalence of 38% in two years, compared with Stanford and Mayer et al. who observed 9.4% (in two years) and 12% (in 34 weeks), respectively [15,16,17]. Our higher prevalence could be explained by including self-reported UTIs by the patient.

Contrary to earlier research, our research showed an increase in IC/BPS complaints after a UTI. Nickel et al. observed that bacteriuria did not appear to be associated with flares of individual symptoms (pain, frequency and urgency) [18]. A possible explanation for this discrepancy is the high percentage of HL+ patients (respectively, 86% versus <13%) in our cohort who may experience more negative effects from a UTI, as these patients may be considered as a more severe subgroup [12]. Furthermore, different questionnaires were applied, focusing on overall experienced symptom severity and perceived burden instead of isolated symptoms. We have to take into account that perceived UTI-like symptoms by the patient can differ from objective measurements. Stanford et al. found that only 18.5% of all UTI-like flare-ups were accompanied by a positive urine culture [16]. However, with respect to burden, patient perception is important. Our findings indicate a significant burden on QoL consistent with several reports that have shown that IC/BPS is related to poorer physical and social functioning and mental health [19,20]. Additionally, frequent UTIs also have a negative impact on patients’ QoL [6].

The second objective of the study was to identify the burden UTIs place on hospital healthcare providers for IC/BPS. The increased number of hospital contacts in the UTI+ group was shown in each specialized center separately. Interestingly, however, there was a significant difference between the specialized centers in the number of visits (data not shown). Patients visited the independent urological clinics three times more often compared with the university medical center. This was in contrast to the telephone contacts, which were four times higher at the university medical center. This did not affect the differences in healthcare consumption between UTI+ and UTI− patients. The differences between centers could depend on different accessibility and management protocols, such as treatment options and involvement of the general practitioner in UTI management. Prior studies tried to identify the burden of IC/BPS on healthcare systems, but stated the difficulty due to the lack of an objective marker for IC/BPS diagnosis [12]. However, studies have shown a substantial economic burden, with annual per-person costs higher than diabetes mellitus, depression, hypertension and asthma. Outpatient and pharmacy expenses comprise a large portion of this [21]. Although no direct cost-analysis was performed, our results show a clear increase in clinical contacts in UTI+ IC/BPS patients that are likely to further increase healthcare costs in these patients.

UTIs in IC/BPS patients also have an impact on a societal level, consisting firstly of the high healthcare consumption and secondly of high antibiotic use. Not surprisingly, patients with UTIs received more antibiotic treatments than patients without UTIs. Moreover, over half of the UTI patients used prophylactic antibiotics. More concerning, in the patients without UTIs 20% still used prophylactic antibiotics for all three years, implying there is overtreatment with antibiotics in IC/BPS. Additionally, in 80% of the urine cultures, antibiotic resistance was observed. The global problem of increasing antibiotic resistance leads to fewer (oral) treatment options for infections, impacting individual patients and society as a whole.

A limitation of the patient file study is that the time of analysis was during the COVID-19 pandemic, which could result in an underestimation of healthcare-seeking behavior; however, substantial differences between UTI− and UTI+ were seen nonetheless. Secondly, the investigated patient files are of healthcare centers that are specialized referral centers, as seen in the high percentage of patients with severe inflammation/Hunner lesions. This might result in selection bias. The survey was available to IC/BPS patients through the Dutch IC/BPS patient association. However, as it was filled-out voluntarily, it might be subject to response bias. Questions were designed to limit survey bias but are not part of a validated questionnaire. Together, this could result in an underrepresentation of IC/BPS patients with mild symptoms.

## 5. Conclusions

This study identifies the burden of UTIs on IC/BPS patients’ QoL and the healthcare system. UTIs are a frequent occurrence in IC/BPS patients and cause major diagnostic delay, increase the perceived IC/BPS symptom severity and lead to more anxiety. UTIs lead to an experienced and objectified increase in healthcare consumption. These results confirm that there is a need to thoroughly address UTI management in IC/BPS guidelines. Clear diagnostic and treatment tools should be defined to minimize over- and undertreatment in order to improve QoL and healthcare management.

## Figures and Tables

**Table 1 healthcare-11-02761-t001:** Demographics of the study population.

A. Data from surveyed patient cohort							
	All Patients (N = 198)		UTI Group (N = 108)		No UTI (N = 83)		
	%	95%-CI	%	95%-CI	%	95%-CI	*p* Values
Sex							***p* = 0.013**
Male	11	[7–16]	5 **	[2–11]	18 **	[10–28]	
Female	89	[84–94]	95	[89–98]	82	[72–90]	
Age							*p* = 0.210
19–40 years	8	[5–13]	9	[5–17]	8	[3–16]	
41–60 years	32	[26–40]	35	[26–45]	27	[17–38]	
61–80 years	54	[47–62]	49	[39–59]	63	[52–74]	
>81 years	5	[2–9]	7	[3–13]	3	[3–9]	
Cystoscopy							*p* = 0.239
Hunner lesions	46	[39–54]	50	[39–60]	43	[32–54]	
Inflammation	44	[36–51]	42	[32–52]	45	[34–57]	
No abnormalities	3	[1–7]	2	[0–7]	5	[1–12]	
Symptoms before diagnosis							*p* = 0.411
0–2 years	47	[40–54]	41	[32–51]	54	[43–65]	
3–5 years	23	[17–30]	24	[17–34]	20	[12–30]	
6–10 years	9	[5–14]	8	[4–15]	10	[4–19]	
11–20 years	12	[8–18]	14	[8–22]	10	[4–19]	
>20 years	9	[6–14]	12	[7–20]	6	[2–14]	
**B. Data from patient file study**							
	**All Patients (N = 100)**		**UTI Group (N = 53)**		**No UTI** **(N = 47)**		
	**%**	**95%-CI**	**%**	**95%-CI**	**%**	**95%-CI**	***p* Values**
Sex							***p* = 0.027**
Male	7	[3–14]	2 **	[1–10]	13 **	[5–26]	
Female	93	[86–97]	98	[90–100]	87	[74–95]	
Age							*p* = 0.806
Median (IQR)	63 (27)		61 (29)		64 (26)		
Cystoscopy							*p* = 0.067
Hunner lesions	84	[75–90]	88	[77–96]	79	[64–89]	
Inflammation	12	[6–20]	12	[4–23]	13	[5–26]	
No abnormalities	4	[1–10]	0	[0–7]	9	[2–20]	

IQR: interquartile range. Percentages are rounded to the nearest whole number [95%-confidence interval]. ** Significances are between UTI+ and UTI− groups, *p*-values in column, bold *p-*values are ≤0.05.

**Table 2 healthcare-11-02761-t002:** Differences in patients’ perspectives from patient survey cohort (UTI+/UTI−).

	UTI Group (N = 108)		No UTI (N = 83)		
	%	95%-CI	%	95%-CI	*p* Values
GP confusing IC/BPS for UTI	84 **	[76–91]	54 **	[43–65]	***p*** **≤ 0.001**
Diagnosis delay by GP due to confusion					-
0–2 years	55	[43–66]	69	[53–82]	
2–5 years	18	[10–28]	18	[8–32]	
6–10 years	13	[6–22]	7	[1–18]	
>10 years	15	[8–25]	7	[1–18]	
Therapeutic antibiotic courses ^1^			-	-	-
0	30	[21–39]			
1–5	49	[39–59]			
>6	21	[14–30]			
Use of antibiotic prophylaxis ^1^	50	[40–59]	-	-	-
Antibiotic resistance in urine cultures ^1^			-	-	-
Yes	32	[23–41]			
Unclear	32	[24–42]			
Fear of a new UTI					***p*** **≤ 0.001**
Yes	70 **	[61–79]	31 **	[21–42]	
Worsening of IC/BPS symptoms after a UTI					***p*** **≤ 0.001**
Yes	60 **	[50–69]	17 **	[9–27]	
Increased healthcare (for IC/BPS) consumption during or after UTI					***p*** **≤ 0.001**
Yes	47 **	[37–57]	13 **	[6–22]	

GP: general practitioner, IC/BPS: interstitial cystitis/bladder pain syndrome, UTI: urinary tract infection. Percentages are rounded to the nearest whole number [95%-confidence interval]. ^1^ Data had been only collected from patients with UTI. ** Significance between the groups UTI and no UTI: *p-*value in column, bold *p-*values are ≤0.05.

**Table 3 healthcare-11-02761-t003:** Differences in healthcare consumption by UTI+ and UTI− patients.

	UTI Group (N = 53)		No UTI (N = 47)		
	Median	IQR [Range]	Median	IQR [Range]	*p*-Values
Nr of UTIs since January 2020	2	2 (1–9)	-	-	-
Nr of digital contacts	0 *	0 [0–2]	0 *	0 [0–7]	***p* ≤ 0.05**
Nr of telephone contacts	3 *	7 [0–36]	1 *	3 [0–23]	***p* ≤ 0.05**
Nr of physical contacts	15 *	17 [1–65]	9*	12 [0–43]	***p* ≤ 0.05**
Total nr of contacts	18 *	24 [4–65]	11 *	15 [1–46]	***p* ≤ 0.05**
Nr of antibiotic therapies	2 **	2 [0–10]	0 **	0 [0–10]	***p* ≤ 0.001**
	**%**	**95%-CI**	**%**	**95%-CI**	
Resistance					
% of all positive urine cultures	80	[56–94]	-	-	-
Use of antibiotic prophylaxis					
At one moment during 2020–2022	75 **	[62–86]	38 **	[25–54]	***p* ≤ 0.001**
Entire study period 2020–2022	51		21		
Use of GAG therapy					*p* = 0.124
At one moment during 2020–2022	85	[72–93]	72	[57–84]	
Entire study period 2020–2022	49		55		
Use of cranberry	37	[25–52]	21	[11–36]	*p* = 0.073

Nr: number, UTIs: urinary tract infections, %: percentages, GAG therapy: glycosaminoglycan therapy. Percentages are rounded to the nearest whole number [95%-confidence interval]. *, ** significant differences between patients with and without UTIs: *p-*value in column, bold *p-*values are ≤0.05.

## Data Availability

The data presented in this study are available on request from the corresponding author. The data are not publicly available due to the fact that the participants of the survey and the patients from the patient files were not informed about data sharing (not included in ‘Geen Bezwaar’ policy).

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
