# Peer review of "The Burden of Urinary Tract Infections on Quality of Life and Healthcare in Patients with Interstitial Cystitis"

_healthcare, 2023, doi:10.3390/healthcare11202761_

Round 1

Reviewer 1 Report

Abstract

·      The abstract should be better aligned as a summary of the manuscript overall, including aim, the design, etc.

·       The methods is too limited for an abstract

Introduction

·       Ln52: Consider to not link impaired membrane barrier as a cause’ of UTI; the cause would be infection, with increased susceptibility or risk because of the impaired barrier.

·       Ln48-49: Consider to rephrase to read a little better

·       Ln61: Use the abbreviation for UTI here too

·       Ln63: Please provide a clear aim of the study, and not ‘purpose’; Please align with the aim provided in Ln277, and also align this to the aim in the abstract (Ln 19-20).

Methods

·       Please include an opening paragraph that includes an overview of the design and the study type (i.e. it appears cross-sectional and/or case-controlled study with a retrospective component).

·       Please include the names of the 4 healthcare centres: i.e. more clear information on the study ‘site(s)’ is needed

·       Please include the ethical clearance number and name of committee that provided this approval in the opening paragraph; Project registration provided in Ln 91-92 for the ‘retrospective’ part of the study, and not clear if it is for all components.

·       Surely ethical clearance with informed consent is needed for the cross-sectional component of this study? It can’t be possible that patient is invited to give data to a study without informed consent; For the retrospective component, this may be waivered depending on the nature of the data, and seems to align to the ‘Geen Bezwaar’ policy cited. However, if a patient is asked to give new data not previously collected for the intention of a research study, this must have ethical clearance.

·       Ln71: Please define further what is meant by ‘retrospective’ questionnaire, as this appears a cross-sectional portion of the study. This can be further clarified please to not confuse the nature of the data collection tool vs the type of quantitative study design used here

·       Please detail how participants were recruited for the cross-sectional component

·       Provide the dates for the cross-sectional component of the study; when was participants recruited and when was data collected (include month and year)? Please also include month with the years (Ln91: 2020 – 2022) for the retrospective component

·       Please include a clear inclusion and exclusion criteria for the cohort selection for the cross-sectional component, and not just the one exclusion criteria. The cohort needs to be more clearly defined.

·       Is there an overlap in participant information from the cross-sectional study (n = 198) and the patient files in the retrospective part of the study (n = 100)

·       Please justify the decision to use n = 100 patient files? Was this based on a sample size calculation?  

·       Recommended to have the statistical analysis at the end for all data in a separate subsection

Results

·       Please include the group comparison in the written text

·       The burden on healthcare component needs more clarity in the methods, results, discussion and conclusion.

Some improvement in syntax and flow, as well as structure of the manuscript, is recommended. These have generally not been itemised in the recommendations. 

Reviewer 2 Report

I have reviewed the study titled ''The Burden of Urinary Tract Infections on Quality of Life and Healthcare in Patients with Interstitial Cystitis''.

The study has a strong hypothesis and evaluates an aspect that has not been adequately analyzed in the literature. In this context, it will contribute to the literature, but some revisions are needed;

1.However, little attention is given to this in guidelines. This sentence should be written more professionally.

2.Methods: should be detailed; methodological type of study? study time interval? Which parameters were analyzed? which standard scorings were evaluated.

3.Results: UTIs..sentences should not be started with an abbreviation.

4.This study identifies a burden of UTIs on IC/BPS patients' quality of life and their healthcare system. This sentence is unnecessary.

5.Materials and Methods: This section can be written with subtitles such as ethical statements, study design, and statistical analysis.

6. Materials and Methods: A simple flow diagram suitable for the study should be made.

7.Results: The results section was written as a one-to-one explanation of the tables, with repetition errors. Insignificant analyses that will not affect the result should be removed.

8. Table 1:  should be made in two parts, and some results should be given with graphics (visual richness should be provided).

9. Table 2: The relevant p values should be shown in the column of the table.

10.As the last paragraph of the discussion, the limitations and strengths of the study should be mentioned.

11. References: at least 2-3 references dated 2023 can be added or replaced with appropriate existing ones

 Minor editing of English language required

Round 2

Reviewer 1 Report

This is much improved presentation of the manuscript. Some concerns on the format of some tables and text is there, but I think this is maybe part of the submission and file conversion process maybe. I am satisfied with the response to comments. 

The English is suitable for publication